# Antimicrobial Resistance Transmission of Multidrug-Resistant Bacteria in Hydroponic Farming Components

**DOI:** 10.3390/pathogens14111134

**Published:** 2025-11-08

**Authors:** Qian Zhang, Ye Htut Zwe, Daisuke Sano, Dan Li

**Affiliations:** 1Department of Food Science & Technology, National University of Singapore, 2 Science Drive 2, Singapore 117542, Singapore; qzhang.11@u.nus.edu (Q.Z.); zwe_ye_htut@sfa.gov.sg (Y.H.Z.); 2National Centre for Food Science, Singapore Food Agency, 7 International Business Park, Singapore 609919, Singapore; 3Department of Civil and Environmental Engineering, Graduate School of Engineering, Tohoku University, Aoba 6-6-06, Aramaki, Aoba-ku, Sendai 980-8579, Miyagi, Japan; daisuke.sano.e1@tohoku.ac.jp

**Keywords:** antimicrobial resistance, conjugative plasmid transfer, hydroponic farming, *Salmonella*, UV

## Abstract

Hydroponic farming offers sustainability benefits, but its microbial safety remains a concern, particularly regarding antimicrobial resistance (AMR) transmission. This study evaluated the potential for conjugative plasmid transfer of multidrug-resistant bacteria in hydroponic systems, using *Salmonella* Saintpaul B23 as a donor and various *Escherichia coli* strains and a self-isolated *Salmonella* strain from a hydroponic system as recipients. The tested bacteria are human enteric bacteria and may have a chance of being introduced into hydroponic systems. The transconjugation assay was conducted in hydroponic solutions and on different hydroponic components. Results revealed that hydroponic solutions and plant substrates could support significant transconjugation (>4 log CFU transconjugants detected in per mL hydroponic solution and >4 log CFU transconjugants detected in per g plant substrates), while facility surfaces showed minimal transfer (<1 log CFU transconjugants detected on per cm^2^ surface). UV irradiation reduced plasmid transfer rates significantly (*p* < 0.05), suggesting its potential as a mitigation strategy, though proper implementation is critical. Antibiotic residues at sub-minimum inhibitory concentrations exhibited varying effects on AMR propagation, with gentamicin and chloramphenicol unexpectedly reducing transconjugants. These findings highlight the complex dynamics of AMR transmission in hydroponics and underscore the importance of monitoring, UV application, and cautious use of recycled waste to ensure microbial safety and mitigate AMR risks in agricultural production.

## 1. Introduction

Antimicrobial resistance (AMR) is an increasing threat to humans, animals, plants, and the whole environment, and it has been listed among the top 10 global public health threats [1]. The most important reason is that antibiotics are widely distributed in all environments because of their extensive and long-term use [2], particularly antibiotic misuse and overuse, reutilization of sludge [3], animal manure [4], and wastewater. The resistant microbes from healthcare facilities, farms, and pharmaceutical manufacturing are considered reservoirs of antimicrobial resistance genes (ARGs) [5,6]. Countries and regions have facilitated multiple responsible actions in response to this complicated, world-scale issue [7]. WHO [8] suggested that monitoring and surveillance are essential priorities.

Hydroponic farming, where crops are grown without soil in aqueous nutrient solutions, is a practical horticultural approach because of its advantages, such as reduced water and land usage. In general, hydroponic farms do not suffer from fewer biotic or abiotic risks, such as pests and climate change, than conventional farms do [9]. However, the opinions about the microbial safety of hydroponic farms are controversial. Compared with soil-based farms, a variety of pathogens can grow under hydroponic conditions and are more likely to be internalized into plants. Pathogens can quickly spread to neighboring plants through nutrient solutions when even one plant is contaminated [10,11]. Once fresh produce is contaminated by pathogens that invade and cause infection, it is difficult to sanitize [12,13]. Warriner et al. [14] isolated internalized *Escherichia coli* from the surface-sterilized roots of spinach from an artificially contaminated hydroponic farm but failed to do so on a soil-based farm. A quantitative microbial risk assessment study demonstrated potential risks of *Salmonella* spp. infections associated with the consumption of hydroponic vegetables [15].

Currently, an increasing number of recipes for nutrient solutions are being developed for hydroponic growth. In the spirit of conserving and recycling resources, ways to redirect waste from other extrinsic sources as nutrients for hydroponic farms are increasingly being explored. For example, in aquaponics [16], wastewater from a fish farm is directly recirculated into hydroponics systems as nutrients for plants. Similarly, animal manure has been supplemented into hydroponic farms [4,17], and methods for converting domestic wastewater [18] and food waste into a suitable organic nutrient source for hydroponic farming have also been investigated [19]. The existing microbiome originating from animal gut, food, or waste along with the ARGs they harbor and any residual antimicrobials previously administered may be unintentionally introduced into the hydroponic system together with the intended nutrients.

We hypothesized that hydroponic farming facilities might become a hotspot for AMR transmission and this study aims to evaluate the AMR transmission of multidrug-resistant bacteria in different hydroponic farming components, including hydroponic solutions, plant growth substrates, and hydroponic facility surfaces. Ultraviolet (UV) irradiation has been applied to prevent pathogen growth and disease outbreaks on hydroponic farms [20]. Antibiotic residues might be introduced to hydroponic farms via the waste recycling chain. Both UV and antibiotic residues have been demonstrated to have controversial influences on AMR transmission in previous studies [21,22,23,24,25]. Accordingly, we also evaluated the influence of UV and antibiotic residues on conjugative AMR plasmid transfer in hydroponic farming systems.

## 2. Materials and Methods

### 2.1. Bacterial Strains and Antibiotics

The donor evaluated in this study was a multidrug resistance plasmid carrier, *Salmonella* Saintpaul B23, which was isolated from fresh retail chicken meat in Singapore [26]. The tested recipients included *Salmonella* Brunei HW5, a self-isolate from local hydroponic farm water in Singapore [27]; *E. coli* ATCC15597; *E. coli* O157:H7 EDL 933; *E. coli* O157:H7 EDL 931, purchased from the American Type Culture Collection; and *E. coli* O157:H7 C7927, an apple cider isolate obtained from Dr. Kun-Ho Seo at Konkuk University, Republic of Korea [28]. The bacterial strains were reactivated in tryptone soy broth (TSB; Oxoid, Agawam, MA, USA) by incubation at 37 °C overnight for 2 consecutive cycles prior to use in this study.

The stock solutions of ampicillin and rifampicin (Sigma–Aldrich, St. Louis, MO, USA) were prepared by dissolving them in sterile deionized water and dimethyl sulfoxide, respectively. These stock solutions were passed through 0.2 μm filters (Sartorius, Gottingen, Germany) and added to tryptone soy agar (TSA; OXOID) plates. In a mixed system, the donor was selectively enumerated on TSA plates with 50 μg mL^−1^ ampicillin. The recipients were adapted to rifampicin by growing the original strains in stepwise increasing concentrations of rifampicin up to 200 μg mL^−1^ in TSB (Oxoid). The transconjugant population was enumerated on TSA plates supplemented with 50 μg mL^−1^ ampicillin and 200 μg mL^−1^ rifampicin.

### 2.2. Antimicrobial Susceptibility Testing

A single colony was picked from a 24-h incubated plate and transferred into 10 mL of phosphate-buffered saline (PBS) to achieve a bacterial suspension of 0.5 McFarland standard turbidity. The bacterial suspension was streaked across Mueller–Hinton agar (MHA) (OXOID) with a sterile cotton swab (Biomedia, Singapore) three times. The antibiotic disks (Table 1; OXOID) were placed onto the agar via sterile forceps, and the plates were incubated at 37 °C for 18–20 h. The diameters of the inhibition zones were measured, and the results were interpreted according to the breakpoints described by the Clinical and Laboratory Standards Institute (CLSI) M100 breakpoint values.

### 2.3. Conjugation Assays

Hydroponic solutions A and B (POWER-GRO, Singapore) were filtered with a nylon membrane filter with a 0.22 μm pore size (Merck, London, UK) and mixed at a ratio of 1:1, and sterile deionized water (DI water) was added to reach an electrical conductivity value of 2.0 mS cm^−1^, as measured by an EC meter (LAQUAtwin EC-33, HORIBA, Kyoto, Japan). The 10% broth was prepared by adding 10 mL of sterile TSB to 90 mL of sterile deionized water. Freshly reactivated bacterial suspensions were washed with PBS and diluted with hydroponic solution or 10% broth to achieve 5 × 10^7^ CFU mL^−1^ bacterial working solutions. A total of 0.1 mL of working suspensions from the donor and recipient were mixed with 1.8 mL of hydroponic solution or 10% broth, and the mixture was incubated at 30 °C for 24 h. The mixture was serially diluted and plated onto selective agar plates as described above.

Lightweight expanded clay aggregate (LECA) is a commonly used plant growth substrate in hydroponic growth systems. LECAs were autoclaved and air-dried in a biological safety cabinet overnight. A total of 20 mL of bacterial working solution (10 mL of donor + 10 mL of recipient) was added to a 50 mL Falcon tube with 10 g of sterile LECA. After incubation at 30 °C for 24 h, the LECAs were transferred into sterile blue-cap bottles, washed with 20 mL of PBS twice and crushed with a sterile stainless-steel hammer. After the mixture was crushed, 10 mL of PBS was added. The bottle was gently shaken, and the suspension was then serially diluted and plated onto selective agar plates as described above.

Polyvinyl chloride (PVC) is a common material used to construct hydroponic systems. The PVC materials were purchased from a local hardware store and were cut into 3.6 cm × 1.8 cm coupons. The coupons were autoclaved and air-dried in a biological safety cabinet overnight. Each coupon was added to 20 mL of bacterial working solution (10 mL of donor + 10 mL of recipient) and incubated at 30 °C for 72 h. For bacterial enumeration, each coupon was transferred to a Petri dish with 10 mL of PBS and scraped with a sterile cell scraper (Biomedia) for 1 min on each surface. The mixture was subsequently transferred into a 15 mL Falcon tube for serial dilution and plating onto selective agar plates as described above.

All experiments were independently performed at least three times.

### 2.4. Influence of UV Light on Conjugation

Bacterial working solutions (0.1 mL of *S.* Saintpaul B23 as a donor + 0.1 mL of *E. coli* 15597 as a recipient) were mixed with 1.8 mL of hydroponic solution in a six-well plate. The UV lamp (UV-30A) was warmed for 10 min before being applied to the samples, and the irradiance value was monitored by an optical radiometer (MS-100, UVP, Cambridge, UK). After reaching UV doses of 0, 5, 10, and 15 mJ cm^2 −1^, the mixtures were transferred into 15 mL Falcon tubes and incubated at 30 °C for 24 h before being serially diluted and plated onto selective agar plates as described above.

### 2.5. Influence of Antibiotics on the Conjugation

Minimum inhibitory concentration (MIC) assays of the bacteria toward multiple antibiotics were performed via the broth microdilution method following CLSI guidelines. To each well of a 96-well plate, 0.1 mL of bacterial suspension with 0.5 McFarland standard turbidity and 0.1 mL of antibiotic at a certain concentration were added. The positive control was prepared by inoculating bacteria without antibiotics. A well containing 0.2 mL of sterile DI water was used as the negative control. The plates were incubated at 37 °C for 16~20 h. The OD_595_ value of each well was measured via a Multiskan FC microplate photometer (Thermo Fisher Scientific, Shanghai, China), and the MIC values were determined.

Bacterial working solutions (0.1 mL of *S.* Saintpaul B23 as a donor + 0.1 mL of *E. coli* 15597 as a recipient) were mixed with 1.8 mL of hydroponic solution together with tetracycline, gentamicin, or chloramphenicol at their sub-MICs (0.5 MIC) and incubated at 30 °C for 24 h before being serially diluted and plated onto selective agar plates as described above.

### 2.6. Data Analysis

Statistical analyses were performed via SPSS software for Windows, version 26. An unpaired *t* test was used for data with two groups, and one-way analysis of variance was used for data with more than two groups. Differences were considered significant when *p* was <0.05.

## 3. Results and Discussion

### 3.1. Selection of Donor and Recipient

Conjugative plasmid transfer is a principal contributor to gene exchange among bacteria [29,30]. Especially for Gram-negative bacteria, the acquisition of AMR genes by *Enterobacteriaceae* is highly related to the transport of large (40 kb~200 kb) conjugative plasmids and smaller mobilizable plasmids (<10 kb) [31].

*S.* Saintpaul B23 was isolated from fresh retail chicken meat in Singapore and is a multidrug resistance plasmid carrier [26]. It harbors four types of replicons, IncHI1A, IncHI1B, IncFIA, and IncN, and the other regions of pSGB23 contain multiple plasmid maintenance and additive systems, which may indicate that it has a broad host [32]. *S.* Saintpaul B23 was thus selected as the donor to evaluate transconjugation in this study.

*Enterobacteriaceae* are commonly present in fresh produce. Many microbes belonging to the *Enterobacteriaceae* family are known human pathogens or opportunistic pathogens, including *Salmonella* and pathogenic *E. coli*. Generic *E. coli* is generally used as an indicator of possible fecal contamination in fresh produce. In addition, evidence has shown that *E. coli* is involved in horizontal gene transfer events rather frequently, regardless of whether it is interspecies or intraspecies. For example, the One Health Antimicrobial Resistance Working Group [33] reported that the incidence density per 10,000 inpatient days for ceftriaxone- and ciprofloxacin-resistant *E. coli* was 23.4 and 36.2, respectively, which are much greater than those of other antibiotic-resistant bacteria, including methicillin-resistant *Staphylococcus aureus* (10.9) and ciprofloxacin-resistant *Klebsiella pneumoniae* (13.0). Therefore, self-isolated *Salmonella* Brunei HW5 from local hydroponic farm water in Singapore [27], three pathogenic *E. coli* strains (*E. coli* O157:H7 EDL 933, *E. coli* O157:H7 EDL 931, and *E. coli* O157:H7 C7927), and the generic *E. coli* ATCC15597 were tested as recipients in this study.

Before conducting the conjugation tests, all of the recipients were tested against the multiple antibiotics listed in Table 1 and confirmed to be pansusceptible to all the tested antimicrobials. This result indicated that the recipients did not carry a plasmid containing the same replicon as the plasmid carried by the donor; thus, receiving the plasmid from the donor would be a distinguishable phenotype.

In addition, it is worth mentioning that there is no harmonization for the quantitative evaluation of bacterial transconjugation. Researchers have created a number of different experimental set-ups using liquid cultures, filters, or agar plates and various terminologies, including conjugation frequency, transfer frequency, recombinant yield, transfer rate, etc., each with a different calculation method. The initial ratio of donors to recipients, temperature, and shaking during the conjugation assay also influence the results [34]. In this study, we inoculated comparable concentrations of donors and recipients into the systems, incubated them at 30 °C to simulate the ambient temperature for hydroponic agriculture in Singapore, and presented the enumeration of the donors, recipients, and transconjugants on selective agar plates after incubation.

As shown in Table 2, the donor *S.* Saintpaul B23 was more competitive than all of the tested recipients, showing higher population levels after incubation than the recipients. The transconjugation between the donor and the different recipients varied; *E. coli* ATCC15597 generated the highest number of transconjugants (6.5 × 10^4^ CFU mL^−1^), and *E. coli* O157:H7 C7927 presented the lowest number of transconjugants (<10 CFU mL^−1^) (Table 2). In the following evaluations, *E. coli* ATCC15597 was selected as the recipient.

### 3.2. Transconjugation with Hydroponic Elements

Our previous study suggested that *Salmonella* could persist in a hydroponic farming environment throughout 6 weeks of lettuce growth, including in the nutrient solution circulating in the hydroponic system, on the inner surface of the hydroponic facility and in the growth substrate of LECA plants. In particular, the plant growth substrate LECA was found to be a niche for *Salmonella* with high population levels, possibly due to its porous structure [35]. In this study, the transconjugation of these three elements in hydroponic systems was tested. As shown in Figure 1, comparable levels of transconjugants were detected from both the nutrient solution (4.8 ± 0.1 log CFU mL^−1^, Figure 1A) and the plant growth substrate LECA in the hydroponic solution (4.2 ± 0.1 log CFU g^−1^, Figure 1B), whereas no transconjugant was detected from the PVC surfaces in the hydroponic solution (<1 log CFU cm^2 −1^, Figure 1C).

Owing to nutrient solution recirculation, hydroponic farms are highly prone to microbial accumulation, especially in the form of biofilms [36]. It is believed that biofilms in aquatic environments are hotbeds for the horizontal gene transfer of ARGs [37,38,39,40]. Compared with the planktonic solution, the biofilm provides a stable environment for the direct contact of donors and recipients. Angles et al. [41] reported significantly greater plasmid transfer between *Vibrio* spp. strains in biofilms attached to glass beads than in those in biofilms in the aqueous phase. Similarly, Savage et al. [42] reported a dramatic increase in plasmid transfer in *Staphylococcus aureus* biofilms. Our results in this study contrast with these reports. This is possibly due to the lower bacterial densities on the PVC surfaces (7.0 ± 0.0 log CFU cm^2 −1^ of the donor, 6.0 ± 0.1 log CFU cm^2 −1^ of the recipient, Figure 1C) than in the nutrient solution (8.7 ± 0.1 log CFU mL^−1^ of the donor, 7.7 ± 0.1 log CFU mL^−1^ of the recipient, Figure 1A) and the plant growth substrate LECA (8.7 ± 0.1 log CFU g^−1^ of the donor, 7.8 ± 0.1 log CFU g^−1^ of the recipient, Figure 1B). Indeed, as reported by Tham et al. [13], the biofilm-forming ability of all five tested *Salmonella* strains was much lower than that of a number of environmental strains isolated from the surface of a hydroponic facility in Singapore.

We also observed that transconjugation occurred more frequently in 10% bacterial culture media than in hydroponic solutions. When tested in liquids and in the plant growth substrate LECA, whereas no significant difference was detected between donors and recipients in 10% broth and in hydroponic solutions (*p* > 0.05), significantly greater numbers of transconjugants were detected in 10% broth than in hydroponic solutions (*p* < 0.05) (Figure 1A,B). When tested on PVC surfaces, there were also no significant differences between the donor and recipient populations on surfaces in 10% broth or in hydroponic solutions (*p* > 0.05). No transconjugants were detected from the PVC surfaces in the hydroponic solution (<1 log CFU cm^2 −1^), whereas transconjugants were detected at 2.7 ± 0.9 log CFU cm^2 −1^ on the surfaces in 10% broth (Figure 1C). The fitness cost is considered a barrier in horizontal gene transfer, and it is known that conjugation is an energy-intensive process requiring cells with significant metabolic activity [43]. However, the fitness cost was found to be easily compensated for by bacterial mutations [44]. In addition, this study was tested in freshly prepared hydroponic solutions with minimal nutrients. In real life, when plants are grown in hydroponic systems, debris and plant exudates are expected to increase organic matter and thus nutrient levels in hydroponic solutions, potentially increasing the chance of horizontal gene transfer.

### 3.3. Influence of UV Light on Transconjugation

UV light is widely used in agricultural production to ensure microbial safety [45]. However, owing to the poor penetration of UV light and the various installation and management practices, this treatment has been reported to have inconsistent antimicrobial effectiveness in hydroponic systems [46,47]. As UV treatment generally causes a partial decrease in microbial loads, the residual microbiome typically quickly regrows to the original level and is sometimes associated with the emergence of new antimicrobial resistance [21,48]. As shown in Figure 2, the susceptibilities of the donor *S.* Saintpaul B23 and the recipient *E. coli* ATCC15597 to low-dose UV treatment were comparable (*p* > 0.05). Although dose-dependent reductions were obtained after UV treatment (Figure 2), both the donor *S.* Saintpaul B23 and the recipient *E. coli* ATCC15597 grew to their original levels in the hydroponic solutions after 24 h of incubation at 30 °C (*p* > 0.05; Figure 3A). Moreover, the transconjugant numbers in the 5, 10, and 15 mJ cm^2 −1^ UV-treated groups were significantly lower than those in the untreated control group (*p* < 0.05; Figure 3A). This result is consistent with previous reports demonstrating that UV irradiation reduces the horizontal transfer of conjugative plasmids in drinking water systems, likely because UV exposure compromises bacterial viability and induces DNA damage responses that shift cellular priorities toward repair rather than conjugation [49]. These findings reveal a previously underreported benefit of applying UV in hydroponic systems.

### 3.4. Influence of Antibiotic Residues on Transconjugation

Antibiotic residues have been reported to contribute to AMR transmission, as they provide selective pressure [50,51,52,53]. As shown in Table 3, the influence of tetracycline, gentamicin, or chloramphenicol on transconjugation at sub-MICs (0.5 MIC) was tested. After incubation at 30 °C for 24 h, no significant influence was observed after treatment with 2 μg mL^−1^ tetracycline (*p* > 0.05; Figure 3B). In contrast to our hypothesis, gentamicin at 1 μg mL^−1^ and chloramphenicol at 4 μg mL^−1^ induced significantly lower transconjugant numbers than did the control (*p* < 0.05; Figure 3B). The mechanism remains unclear and is worthwhile to be explored in the future.

## 4. Limitations

This study has several limitations. The experiments were conducted only under controlled laboratory conditions and over a short duration. In real-world hydroponic farms, long-term adaptation of the microbiome within urban farming facilities and biofilm formation may influence the conjugative transfer of AMR plasmids. Moreover, plant exudates in the hydroponic solution and the composition of the microbiome as tested previously [13] are also likely to play important roles. Therefore, future studies should evaluate these findings in real or pilot-scale hydroponic systems to confirm their practical relevance.

## 5. Conclusions

In summary, the results of this study indicated that hydroponic facilities have moderate potential to contribute to the dissemination of AMR. UV light might be able to mitigate conjugative AMR plasmid transfer in hydroponic farming systems, but the installation and management of UV light still need a good plan. Finally, vigilant consideration should be given to the use of waste as a fertilizer source in hydroponic farms, and there is a need for increased measures to control the propagation of AMR.

## Figures and Tables

**Figure 1 pathogens-14-01134-f001:**
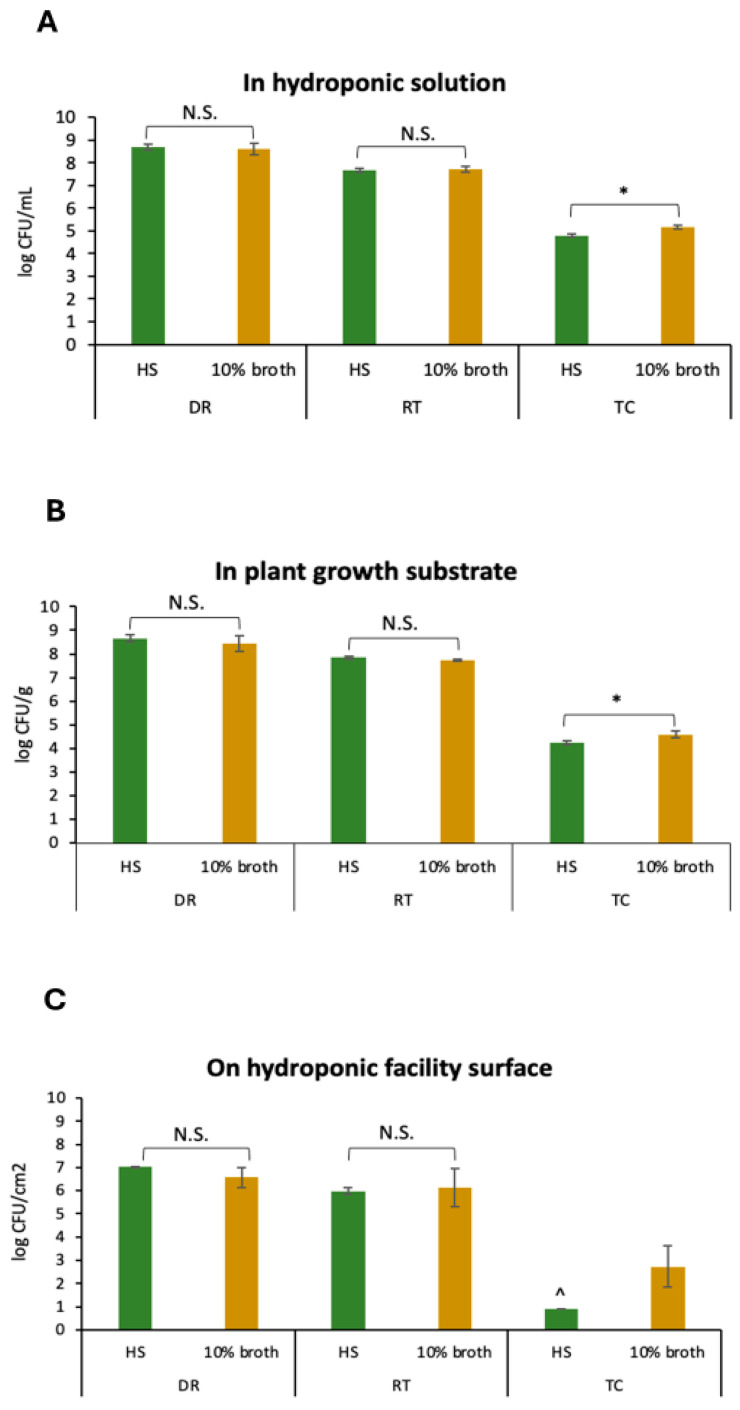
Enumeration of *S.* Saintpaul B23 as a donor (DR), *E. coli* ATCC15597 (RT), and the transconjugants (TC) in hydroponic solution (**A**), in plant growth substrates (**B**), and on hydroponic facility surfaces (**C**) in the conjugation assays. Each column represents the average of triplicates, and each error bar indicates the standard deviation. * denotes a significant difference between the results in hydroponic solution (HS) and 10% broth (*p* < 0.05 by unpaired *t* test); N.S. denotes no significance. ^ denotes values below the detection limit (1 log CFU/cm^2^).

**Figure 2 pathogens-14-01134-f002:**
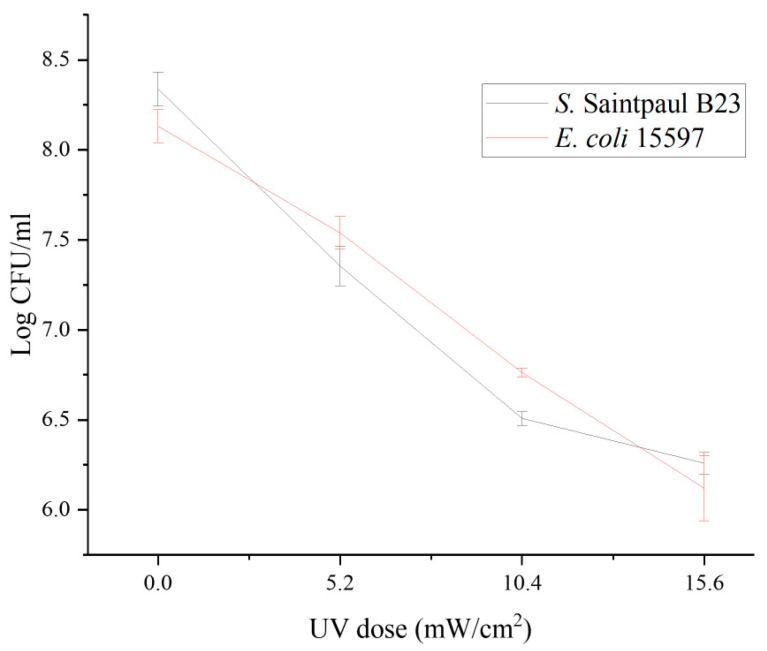
UV inactivation of *S.* Saintpaul B23 and *E. coli* ATCC15597. Each data point represents the average of triplicates, and each error bar indicates the standard deviation.

**Figure 3 pathogens-14-01134-f003:**
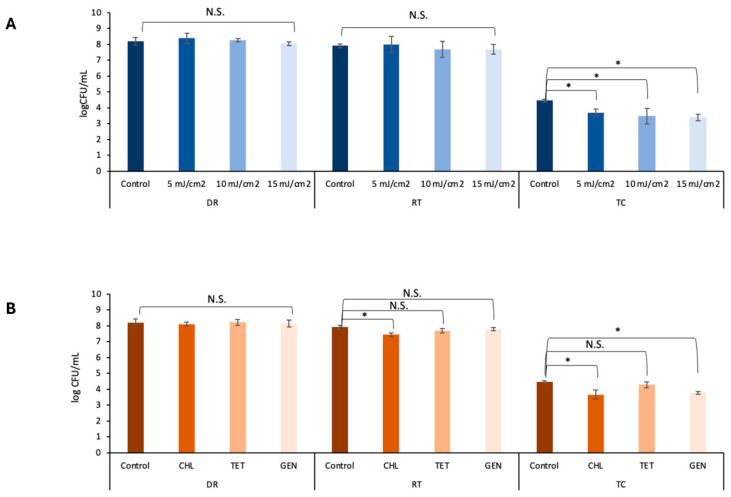
Enumeration of *S.* Saintpaul B23 as a donor (DR), *E. coli* ATCC15597 (RT), and the transconjugants (TC) in hydroponic solution after different doses of UV radiation (**A**) and after different antibiotic treatments at sub-MIC doses (**B**). TET denotes tetracycline, GEN denotes gentamicin, and CHL denotes chloramphenicol. Each column represents the average of triplicates, and each error bar indicates the standard deviation. * denotes a significant difference between the results in hydroponic solution (HS) and 10% broth (*p* < 0.05 by one-way analysis of variance); N.S. denotes no significance.

**Table 1 pathogens-14-01134-t001:** List of antimicrobial discs used, and their category and concentration.

Disk	Category	Conc. (μg/disc)
Ampicillin	Penicillin	10
Ceftriaxone	Cephalosporin	30
Tetracycline	Tetracycline	30
Ciprofloxacin	Quinolone and Fluoroquinolone	5
Nalidixic acid	Quinolone and Fluoroquinolone	30
Tobramycin	Aminoglycoside	10
Chloramphenicol	Phenicol	30
Fosfomycin	Fosfomycin	200
Imipenem	Carbapenem	10
Sulfamethoxazole and trimethoprim	Folate pathway inhibitor	23.75 and 1.25

**Table 2 pathogens-14-01134-t002:** Conjugation of pSGB23 plasmid from *S.* Saintpaul B23 to different recipients in TSB.

DR	RT	Enumeration (CFU mL^−1^)		
		Donor	Recipient	Transconjugant
*S.* Saintpaul B23	*E. coli* ATCC15597	$9.1\times 10^{8}$	$6.2\times 10^{8}$	$6.5\times 10^{4}$
	*E. coli* O157:H7 EDL933	$1.1\times 10^{9}$	$3.9\times 10^{8}$	$9.7\times 10^{3}$
	*E. coli* O157:H7 EDL931	$2.1\times 10^{9}$	$2.3\times 10^{8}$	$3.8\times 10^{3}$
	*E. coli* O157:H7 C7927	$8.5\times 10^{8}$	$4.7\times 10^{8}$	<10
	*S.* Brunei HW5	$1.0\times 10^{9}$	$5.2\times 10^{8}$	$3.2\times 10^{3}$

**Table 3 pathogens-14-01134-t003:** MIC values (μg mL^−1^) of tested bacteria towards relevant antibiotics and their susceptibility status. Abbreviation: R, resistant; S, susceptible.

	*S.* Saintpaul B23	*E. coli* 15597
Gentamicin	128 (R)	2 (S)
Tetracycline	256 (R)	4 (S)
Chloramphenicol	64 (R)	8 (S)

## Data Availability

The original contributions presented in this study are included in the article. Further inquiries can be directed to the corresponding author.

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
