# Peer review of "Antimicrobial Resistance Transmission of Multidrug-Resistant Bacteria in Hydroponic Farming Components"

_pathogens, 2025, doi:10.3390/pathogens14111134_

Round 1

Reviewer 1 Report

Comments and Suggestions for Authors

The present study entitled “Antimicrobial Resistance Transmission of Multidrug-Resistant Bacteria in Hydroponic Farming Components” investigates how antimicrobial resistance spreads in hydroponic farming systems, where plants grow in nutrient-rich water instead of soil. Using multidrug-resistant Salmonella Saintpaul and various E. coli and Salmonella strains, the researchers found that resistance genes could transfer easily in hydroponic solutions and plant substrates, but not on facility surfaces. Ultraviolet light reduced gene transfer, suggesting it may help control AMR, while low levels of some antibiotics unexpectedly lowered transfer rates. The findings highlight that hydroponic systems can facilitate AMR transmission and stress the need for proper UV treatment, monitoring, and careful use of recycled materials to ensure food safety.

Query 1: The introduction gives good background information on antimicrobial resistance and hydroponic farming, but it is quite long and includes many global examples. Some parts could be shortened to focus more clearly on the knowledge gap and the reason for this particular study.

Query 2: As I can see, the experiments were done only under laboratory conditions, not in real hydroponic farms. Please write it as a limitation and also mention that future studies should test these findings in real or pilot-scale hydroponic systems to confirm their practical relevance.

Query 3: The study focused mainly on short-term laboratory experiments, observing bacterial interactions over only a few days. This limited duration does not reflect the full crop cycle in hydroponic farming, where conditions such as nutrient changes, biofilm formation, and microbial adaptation may influence antimicrobial resistance over time. As a result, the long-term stability, persistence, or possible accumulation of resistant bacteria in hydroponic systems remains unclear. I’m curious what authors think about it. Please respond.

Query 4: The discussion should include a clear statement of the study’s limitations. Mentioning them, if they exist, would make the work more transparent and help readers better understand the scope and reliability of the results.

I congratulate the authors for their work and I look forward to their answers.

Reviewer 2 Report

Comments and Suggestions for Authors

This paper submitted for review assessed the potential transfer of antimicrobial resistance via conjugative plasmid transfer among multidrug-resistant bacteria under hydroponic agricultural systems. The topic is very relevant, considering that it addresses a rapidly emerging problem with sustainable agriculture and food safety. However, I believe that this work could be improved through the points raised in the attached reviewer comment.

Reviewer 3 Report

Comments and Suggestions for Authors

The study demonstrates the transmission of antimicrobial resistance (AMR) in hydroponic systems through conjugative plasmid transfer. It employs Salmonella saintpaul B23 as a donor and multiple E. coli isolates and one Salmonella isolate as recipients. The study is important for a critical interface between environmental microbiology and food safety. However, I have several queries to be addressed before it goes to the further stages.

  1. The authors should show the plasmid compatibility, or gene expression related to conjugation machinery.
  2. The study was performed in the lab condition using hydroponic systems in sterile conditions. What about the real hydroponic farm where complex microbial communities are existed.
  3. The authors should clearly mention the number of replicates and the control condition. There is no information about it in the manuscript or even in the figure legends.
  4. The authors claimed that UV mitigates the conjugation. What are the evidence of this claim.
  5. I found too many self-citations such as [13], [15], [25], [26], [27], [32], [35]. It means the over-dependence on their previous works.
  6. The study concept of AMR transfer in hydroponic environments is quite familiar. There are many similar studies on biofilm-mediated transfer and nutrient solution-based systems exist. What is the novelty of this study?
  7. The authors used many times the words such as “in this study,” “In addition,” “Therefore,” and “As a result” create redundancy throughout the manuscript. Please use some alternative words.
  8. The authors claimed that UV has “previously unknown benefits” (line 275). Please double check this information. I think previous studies have already reported similar finding.

Reviewer 4 Report

Comments and Suggestions for Authors

please see the file.

Round 2

Reviewer 2 Report

Comments and Suggestions for Authors

The manuscript has been improved sufficiently using the reviewer's comments. Before acceptance, there are minor corrections which can help improve the quality of the manuscript aas follows;

Line 70, correct to "facilities"

The conclusion of the study was spent highlighting the limitations. It is suggested that a separate limitation section be included in the manuscript and the conclusion should focus on summarising the manuscript and highlighting the manuscripts contribution to the research area and possibly propose future studies.

Author Response

Line 70, correct to "facilities"

Answer: done.

The conclusion of the study was spent highlighting the limitations. It is suggested that a separate limitation section be included in the manuscript and the conclusion should focus on summarising the manuscript and highlighting the manuscripts contribution to the research area and possibly propose future studies.

Answer: done.

Thanks again for reviewing our work!!

Reviewer 3 Report

Comments and Suggestions for Authors

Thank you for the clarification point by point.

Reviewer 4 Report

Comments and Suggestions for Authors

Please see the file.
